# The contribution of common regulatory and protein-coding *TYR* variants to the genetic architecture of albinism

Vincent Michaud [1,2,7], Eulalie Lasseaux[1,7], David J. Green[3,7], Dave T. Gerrard [3], Claudio Plaisant[1], UK Biobank Eye and Vision Consortium*, Tomas Fitzgerald [4], Ewan Birney [4], Benoît Arveiler [1,2✉], Graeme C. Black [3,5✉] & Panagiotis I. Sergouniotis [3,4,5,6✉]

Genetic diseases have been historically segregated into rare Mendelian disorders and common complex conditions. Large-scale studies using genome sequencing are eroding this distinction and are gradually unmasking the underlying complexity of human traits. Here, we analysed data from the Genomics England 100,000 Genomes Project and from a cohort of 1313 individuals with albinism aiming to gain insights into the genetic architecture of this archetypal rare disorder. We investigated the contribution of protein-coding and regulatory variants both rare and common. We focused on *TYR*, the gene encoding tyrosinase, and found that a high-frequency promoter variant, *TYR* c.−301C>T [rs4547091], modulates the penetrance of a prevalent, albinism-associated missense change, *TYR* c.1205G>A (p.Arg402Gln) [rs1126809]. We also found that homozygosity for a haplotype formed by three common, functionally-relevant variants, *TYR* c.[−301C;575C>A;1205G>A], is associated with a high probability of receiving an albinism diagnosis (OR>82). This genotype is also associated with reduced visual acuity and with increased central retinal thickness in UK Biobank participants. Finally, we report how the combined analysis of rare and common variants can increase diagnostic yield and can help inform genetic counselling in families with albinism.

[1] Department of Medical Genetics, University Hospital of Bordeaux, Bordeaux, France. [2] INSERM U1211, Rare Diseases, Genetics and Metabolism, University of Bordeaux, Bordeaux, France. [3] Division of Evolution, Infection and Genomics, School of Biological Sciences, Faculty of Biology, Medicine and Health, University of Manchester, Manchester, UK. [4] European Molecular Biology Laboratory, European Bioinformatics Institute (EMBL- EBI), Wellcome Genome Campus, Cambridge, UK. [5] Manchester Centre for Genomic Medicine, Saint Mary's Hospital, Manchester University NHS Foundation Trust, Manchester, UK. [6] Manchester Royal Eye Hospital, Manchester University NHS Foundation Trust, Manchester, UK. [7] These authors contributed equally: Vincent Michaud, Eulalie Lasseaux, David J. Green. *A list of authors and their affiliations appears at the end of the paper. ✉email: benoit.arveiler@chu-bordeaux.fr; graeme.black@manchester.ac.uk; panagiotis.sergouniotis@manchester.ac.uk

There is abundant evidence supporting the view that rare genetic diseases are caused by rare, high-impact variants in individual genes[1,2]. However, for most known rare disorders, it is not possible to identify such pathogenic changes in every affected proband, leaving significant diagnostic and knowledge gaps[3–5]. In recent years, the emergence of comprehensive rare disease and population-based resources that link genomic and phenotypic data (e.g. UK Biobank[6], Genomics England 100,000 Genomes Project[7]) has offered unprecedented opportunities for genetic discovery[8–13]. Through integrative analysis of these datasets, we can now achieve line-of-sight for uncovering complex molecular explanations in people with rare disorders who have hitherto remained undiagnosable.

Albinism, a rare recessive condition characterised by decreased ocular pigmentation and altered visual system organisation[14], had a pivotal role in the study of human genetics tracing back to the early 20th century[15,16]. At least 20 genes are now known to be associated with this disorder and the current diagnostic yield of genetic testing in affected cohorts approaches 75%[17–19]. Most people with a molecular diagnosis of albinism carry biallelic variants in *TYR*[17–19]. This gene encodes tyrosinase, the rate-limiting enzyme of melanin biosynthesis[20]. Although individuals with no residual tyrosinase activity have a consistent phenotype with visual impairment and near-total loss of melanin pigment in their eyes, skin and hair, most individuals with albinism fall along a phenotypic continuum with varying degrees of ocular and cutaneous hypopigmentation[14]. Building on recent work[17,21], we sought to increase our understanding of the genetic complexity and clinical heterogeneity of this archetypal disorder.

## Results and Discussion

A cohort of 1208 people with albinism underwent testing of ≤19 albinism-related genes; these individuals were not known to be related and had predominantly European ancestries (Supplementary Data 1). A further 105 probands with albinism were identified in the Genomics England 100,000 Genomes Project dataset[7]. A 'control' cohort of 29,497 unrelated individuals that had no recorded diagnosis or features of albinism was also identified in this resource (Fig. 1, Supplementary Table 1, Methods).

To gain insights into the contribution of common variants to the genetic architecture of albinism, we studied the impact of protein-coding changes that have minor allele frequency [MAF] ≥ 1% and are predicted by a computational algorithm to be functionally relevant (CADD score[22] ≥ 20). For *TYR*, two such variants were identified: c.575C>A (p.Ser192Tyr) [rs1042602] and c.1205G>A (p.Arg402Gln) [rs1126809]. Multiple associations have been recorded for these two changes including with skin/hair pigmentation (for both variants), macular thickness (for c.575 C>A) and iris colour (for c.1205G>A)[23]. Furthermore, each of these changes has been shown to decrease tyrosinase activity in vitro[24–26]. Importantly, there is evidence suggesting that c.1205G>A is acting as a 'hypomorphic' variant, causing a mild form of albinism when in compound heterozygous state with a complete loss-of-function *TYR* change[27]. It is also noted that the MAF of this variant in European populations is around 27% and that multiple unaffected homozygous individuals have been reported (including > 2000 people in the control subset of the Genome Aggregation Database [gnomAD] v2.1.1)[28].

To gain insights into the contribution of regulatory variants, we studied the impact of changes that alter *TYR* regulatory elements (i.e. the *TYR* promoter or ENCODE-listed enhancers)[29] and affect *TYR* gene expression (i.e. they are known *TYR* expression quantitative trait loci [eQTL]). One such variant was identified, c.−301C>T [rs4547091], a foetal retinal pigment epithelium (RPE) selective eQTL[30]. This change is known to alter a binding

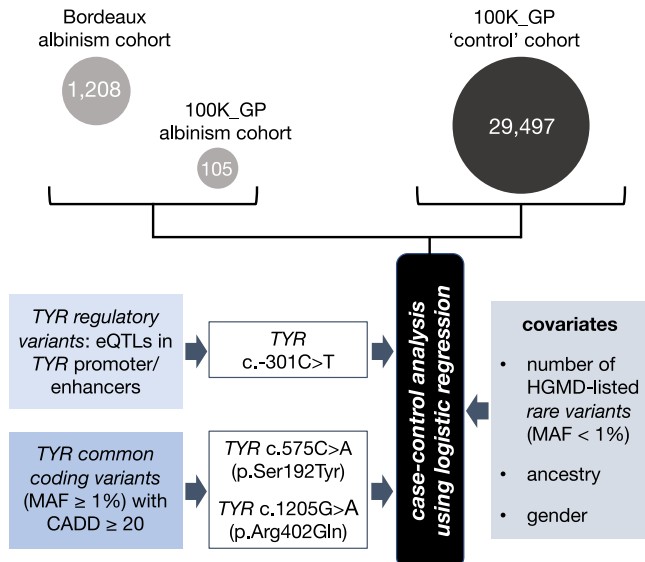

**Fig. 1 Outline of the case–control study design.** A case–control analysis was performed to gain insights into the contribution of protein-coding and regulatory variation (at the common and rare ends of the allele-frequency spectrum) in albinism. The majority of participants in the 'case' cohort (1208/1313) were identified through the database of the University Hospital of Bordeaux Molecular Genetics Laboratory, France. All these probands had at least one key ocular feature of albinism, i.e. nystagmus or prominent foveal hypoplasia. The remaining 105/1313 cases were identified through the Genomics England 100,000 Genomes Project dataset and had a diagnosis of albinism or a phenotype deemed consistent with partial/ocular albinism. The 'control' cohort included 29,497 unrelated individuals from the Genomics England 100,000 Genomes Project dataset, none of whom had a recorded diagnosis of albinism. Genotypes that include selected *TYR* haplotypes in homozygous state were studied. Haplotypes of interest were defined as those formed by combinations of *TYR* variants that are predicted to be functionally relevant; three variants met the pre-determined criteria set for regulatory (*TYR* c.−301C>T) and common protein-coding (*TYR* c.575C>A and c.1205G>A) variants. The associated haplotypic blocks were analysed further using logistic regression (see Methods). 100K_GP Genomics England 100,000 Genomes Project, eQTLs expression quantitative trait loci, MAF minor allele frequency, CADD Combined Annotation Dependent Depletion score, HGMD Human Gene Mutation Database v2021.2. *TYR* variant numbering is based on the transcript with the following identifiers: NM_000372.5 and ENST00000263321.6.

site for the transcription factor *OTX2* in the *TYR* promoter, and the reference allele (c.−301C) has been shown to lead to a remarkable decrease in promoter activity in vitro[31].

Focusing on individual sequence alterations without consideration for variant interactions and/or patterns of linkage disequilibrium can lead to masking of complex underlying mechanisms. To overcome this pitfall, we avoided an independent analysis of each of the *TYR* c.−301C>T, c.575C>A and c.1205G>A changes and instead studied the haplotype blocks that they form. Eight possible haplotypes [$2^3$] and 36 possible haplotype pairs [$2^{3-1} \times (2^3 + 1)$] may be encountered. We focused only on the 8 haplotype pairs that include homozygous alleles (Fig. 2a) for two reasons: (1) in homozygous individuals, the underlying haplotypes can be unambiguously determined, even in cases where segregation/phasing data are unavailable; (2) in autosomal recessive disorders like *TYR*-related albinism, phenotypic abnormalities are the result of the combined effect of two alleles; by analysing only homozygous cases, the effect of a specific haplotype can be isolated and estimated with greater precision.

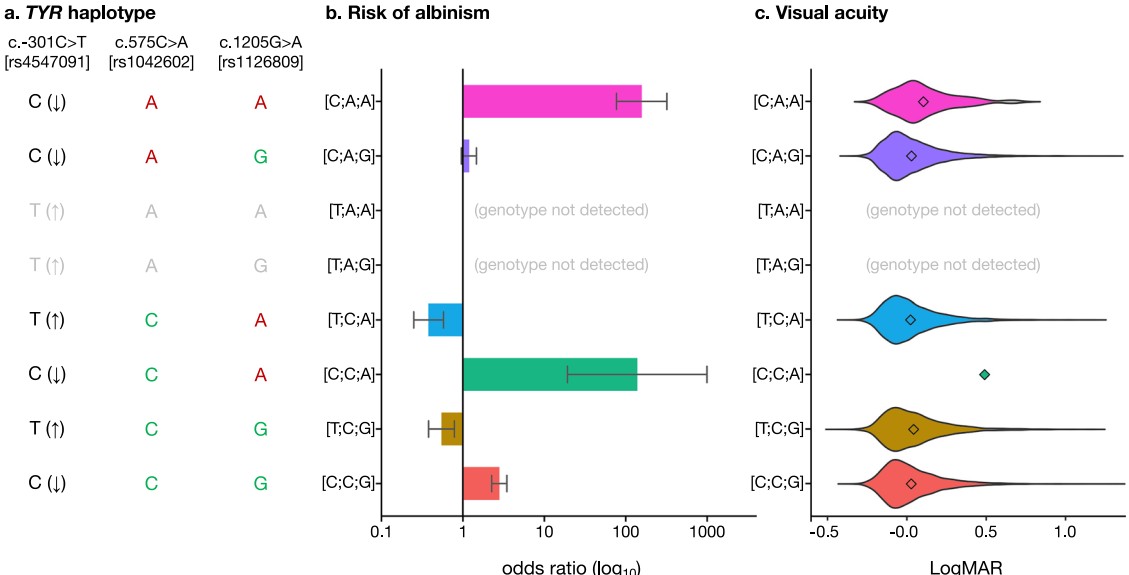

**Fig. 2 Common *TYR* variants form haplotypes that affect risk of albinism and visual performance. a** The *TYR* haplotypes that were studied are shown. The reference allele of the *TYR* c.−301C>T [rs4547091] promoter variant is associated with lower gene expression and is shown as C (↓); the non-reference allele is associated with higher gene expression and is shown as T (↑). The reference alleles of the c.575C>A (p.Ser192Tyr) [rs1042602] and the c.1205G>A (p.Arg402Gln) [rs1126809] missense variants are highlighted in green font while the non-reference alleles are highlighted in red font. As no homozygotes for the *TYR* [−301T;575A;1205A] and [−301T;575A;1205 G] haplotypes were detected, these combinations are highlighted in grey font. **b** Risk of albinism (*i.e.* probability of receiving a diagnosis of albinism) for groups of individuals that carry selected *TYR* haplotypes in homozygous state. The length of each bar chart is proportionate to the value of the point estimate of each odds ratio; the relevant 95% confidence intervals are also shown. A $\log_{10}$ scale with 1 as the reference point is used; it is noted that an odds ratio >1 suggests an increased risk while an odds ratio <1 suggests a decreased risk. Further information including numerical data can be found in Supplementary Table 2. **c** Distribution of visual acuity measurements in UK Biobank participants carrying selected *TYR* haplotypes in homozygous state. Vision near 0.0 LogMAR is considered normal while vision >0.5 LogMAR is considered moderate/severe visual impairment. The Kruskal-Wallis *p* value was $8 \times 10^{-11}$. Further information including numerical data can be found in Supplementary Table 4.

We used Firth regression analysis[32,33] to study how *TYR* haplotypes affect the risk of albinism (i.e. the probability of having a diagnosis of albinism). This increasingly recognised logistic regression approach has been designed to handle small, imbalanced datasets (which are common in studies of rare conditions) and allows for adjustment of key covariates (which is not possible in contingency table methods) (Fig. 1, Methods). The results are shown in Fig. 2b and in Supplementary Table 2. This analysis identified a number of pertinent points that are discussed below.

We found that the penetrance of the 'hypomorphic' *TYR* c.1205G>A variant[27] is modulated by the *TYR* c.−301C>T promoter change. When c.1205G>A is encountered in a homozygous state and in combination with the c.−301C allele of the promoter variant (which reduces *TYR* expression), the risk of albinism is high (OR > 24; see [C;A;A] and [C;C;A] in Fig. 2b). In contrast, homozygosity for c.1205G>A combined with the c.−301T allele (which increases *TYR* expression) has a protective effect (OR < 0.7; see [T;C;A] in Fig. 2b). This observation is in keeping with previous studies suggesting that penetrance can be modified by the joint functional effects of regulatory and protein-coding variants[34]. We here provide a key illustration of this mechanism in the context of a recessively-acting, partial loss-of-function variant.

Alongside this, we found that homozygosity for the *TYR* c.−301C>T promoter variant protects against albinism (OR 0.3–0.7; see [T;C;A] and [T;C;G] in Fig. 2b). Notably, the allele frequency of the protective c.−301T allele, approaches 80% in people of African ancestries and is around 40% in people of European ancestries[28] (see Supplementary Fig. 1 for the geographical distribution of the associated variants/haplotypes). It can be speculated that variation in this *TYR* promoter position partly accounts for the relatively low prevalence of *TYR*-related albinism in people of African ancestries (Supplementary Table 3).

Our findings also highlight that homozygosity for the haplotype formed by the c.−301C allele of the promoter variant (which reduces *TYR* expression) and the non-reference alleles of the two common missense changes, c.575C>A and c.1205G>A, is associated with a significant increase in the risk of albinism (OR > 82; see [C;A;A] in Fig. 2b). Although this haplotype is present in ~1% of people with European ancestries in the 1000 Genomes Project (phase 3)[35], it appears to confer a risk of albinism that is comparable to that of a Mendelian mutation. This conclusion is supported by the findings of three smaller-scale studies that used family-based methods and investigated haplotypes containing the *TYR* c.575C>A and c.1205G>A variants[36–38].

When the *TYR* c.[−301C;575A;1205A] and c.[−301C;575C;1205A] haplotypes (corresponding to [C;A;A] and [C;C;A] in Fig. 2b) were factored in as Mendelian variants in a clinical-grade analysis of the case cohort, the diagnostic yield increased from 57% (692/1208) to 76% (916/1208) (Fig. 3). It is noted that current genetic laboratory pipelines are generally suboptimally set up to identify these complex high-risk haplotypes, especially when filtering is based on the rarity of individual variants.

We subsequently studied the impact of the *TYR* c.[−301C;575A;1205A] haplotype (corresponding to [C;A;A] in Fig. 2b) in UK Biobank participants. We found that people who were homozygous for this haplotype had, on average, reduced visual acuity (mean LogMAR vision 0.10; Kruskal-Wallis *p* value $8 \times 10^{-11}$ with all pairwise comparisons involving [C;A;A] being statistically significant; Fig. 2c and Supplementary Table 4). As visual acuity is a quantitative endophenotype of albinism, this finding provides additional evidence supporting the functional significance of this complex haplotype. A similar trend was noted when central retinal thickness, another albinism endophenotype, was assessed (Kruskal-Wallis *p* value $<2 \times 10^{-16}$; Supplementary Fig. 2 and Supplementary

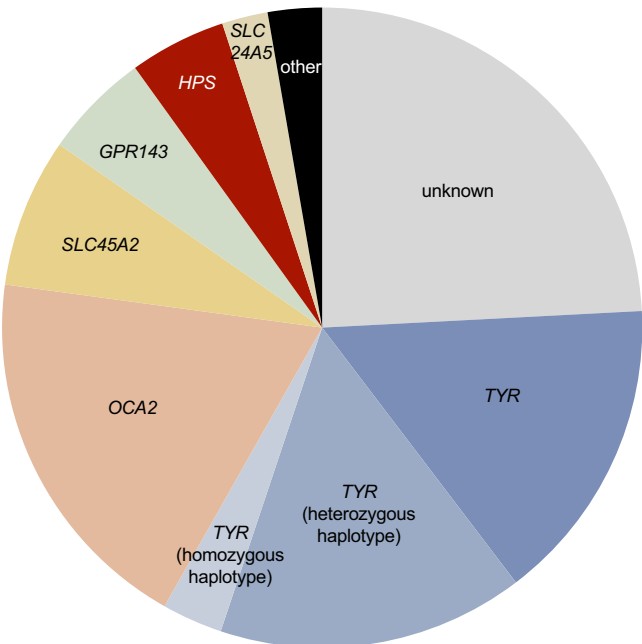

**Fig. 3 High-level molecular diagnoses in 1208 probands from the University Hospital of Bordeaux albinism cohort.** It was not possible to detect a molecular diagnosis in 24% of cases ('unknown' category). The following genes were implicated in the remaining probands: *TYR* (34%), *OCA2* (19%), *SLC45A2* (8%), *GPR143* (5%), Hermansky-Pudlak syndrome (HPS) related genes (5%), *SLC24A5* (2%), other albinism-related genes (3%). A significant subset of cases with *TYR*-related albinism was found/ presumed to carry either the *TYR* c.[−301C;575A;1205A] or the *TYR* c.[−301C;575 C;1205A] haplotype (15% in heterozygous state ['*TYR* heterozygous haplotype' category]; 3% in homozygous state ['*TYR* homozygous haplotype' category]). Further information including a list of all molecular diagnoses can be found in Supplementary Data 1.

Table 5). We expect that future studies analysing visual function and ocular structure in this group of homozygous individuals will provide key insights into the elusive link between RPE melanin synthesis and visual system organisation[39]. Furthermore, we anticipate that the study of cellular models specific to these homozygous cases (e.g. human induced pluripotent stem cell-derived RPE) will advance our understanding of the molecular pathology of albinism.

Lastly, we quantified the risk of albinism associated with combinations of rare and common variants. For each study participant, we estimated two key contributors to an individual's risk. First, we counted the number of rare, presumed Mendelian variants in albinism-related genes; single nucleotide variants that have MAF < 1% and are labelled as disease-causing (DM) in the Human Gene Mutation Database (HGMD) v2021.2[40] were considered. Subsequently, we counted the number of common 'risk genotypes' in *TYR* (i.e. c.−301C, c.575A and/or c.1205A). We found that the presence of >4 common *TYR* risk genotypes confers an increased risk of albinism even in the absence of a rare, HGMD-listed variant (OR > 3.6; Table 1). We also found that, when a single heterozygous HGMD-listed variant co-occurs with >1 common *TYR* risk genotype, the risk of albinism is increased (OR > 4.2 for rare variants in any albinism-related gene, OR > 1.2 for rare variants in *TYR*; Table 1). These observations provide a basis for more precise genetic counselling in families with albinism.

One potential limitation of this study is our inability to stringently match the albinism cases with the unaffected controls, especially in terms of recent ancestry (which can be correlated with skin pigmentation). Although ancestry was included as a parameter in our regression model, this analysis was imperfect as it was not possible to

---

**Table 1 Contribution of different classes of albinism-related variants to disease risk.**

| Combination of common[a] and rare[b] risk genotypes | Odds ratio[c] | 95% confidence interval | p value |
|---|---|---|---|
| 6 common + 0 rare | 391 | 165–982 | 0.000016 |
| 5 common + 0 rare | 7.8 | 3.6–16.5 | <0.00001 |
| 4 common + 0 rare | 3.4 | 1.9–6.1 | 0.000014 |
| 3 common + 0 rare | 1.5 | 0.8–2.6 | 0.17 |
| 2 common + 0 rare | 2.3 | 1.3–4 | 0.002 |
| 1 common + 0 rare | 1.1 | 0.6–1.9 | 0.84 |
| 0 common + 0 rare | 0.1 | 0.1–0.2 | <0.00001 |
| 6 common + 1 rare[d] | 217 | 26.7–2519 | 0.00001 |
| 5 common + 1 rare | 302 | 156–566 | <0.00001 |
| 4 common + 1 rare | 38 | 22–69 | <0.00001 |
| 3 common + 1 rare | 8.7 | 4.9–15.6 | <0.00001 |
| 2 common + 1 rare | 7.5 | 4.2–13.5 | <0.00001 |
| 1 common + 1 rare | 6 | 3.2–11.2 | <0.00001 |
| 0 common + 1 rare | 2.6 | 1.2–5.3 | 0.016 |
| 0 common + 2 rare | 163 | 75–367 | <0.00001 |
| 6 common + 1 rare TYR[d] | 120 | 6–17919 | 0.003 |
| 5 common + 1 rare TYR | 178 | 82–425 | <0.00001 |
| 4 common + 1 rare TYR | 38.6 | 20–80 | <0.00001 |
| 3 common + 1 rare TYR | 5.3 | 2.6–11 | <0.00001 |
| 2 common + 1 rare TYR | 2.4 | 1.2–5.4 | 0.02 |
| 1 common + 1 rare TYR[d] | 0.6 | 0.2–1.9 | 0.43 |
| 0 common + 1 rare TYR[d] | 2.3 | 0.4–8 | 0.28 |
| 0 common + 2 rare TYR[d] | 446 | 100–4270 | <0.00001 |

[a]Presence of each of the following *TYR* variants in one allele was considered as one common risk genotype: c.−301C [rs4547091]; c.575C>A (p.Ser192Tyr) [rs1042602]; c.1205G>A (p.Arg402Gln) [rs1126809]. *TYR* variant numbering is based on the transcript with the following identifiers: NM_000372.5 and ENST00000263321.6.
[b]Autosomal albinism-related genes were inspected and each heterozygous protein-coding change with a minor allele frequency <1% and a 'disease-causing' (DM) label in the Human Gene Mutation Database (HGMD) v2021.2 was considered as one rare risk genotype. Such genotypes in the *TYR* gene are labelled '*rare TYR*'.
[c]Firth regression analysis with gender and ancestry as covariates was used to estimate odds ratios and unadjusted p values; 1313 probands with albinism and 29,497 unrelated controls were included in this analysis (Supplementary Data 1).
[d]Less than 10 individuals were included in the smallest subgroup of these categories, reducing the reliability of the associated findings.

---

reliably assign genetic ancestry to most albinism cases. It is known that inability to fully account for differences in ancestral background between cases and controls can lead to false-positive association signals[41]. We used a combination of orthogonal approaches to evaluate the robustness and generalisability of our findings. First, we used 35 presumed neutral single-nucleotide variants to calculate the genomic inflation factor lambda ($\lambda$GC)[42,43]; $\lambda_{median}$ was found to be 1.04, in keeping with limited confounding by ancestry (Supplementary Table 6). Subsequently, we performed targeted sub-analyses of the available cohorts; the results of three focused case-control studies supported our key findings and increased confidence in the validity of the detected associations (Supplementary Fig. 3 and Supplementary Tables 7–9).

In conclusion, we have shown that a significant proportion of albinism risk arises from genetic susceptibility linked to common variants. Furthermore, our findings suggest that rare and common protein-coding variation in *TYR* should be considered in the context of regulatory haplotypes. The concepts discussed here are likely to be relevant to the understanding of other rare disorders,

and haplotype-based approaches are expected to narrow the diagnostic gap for significant numbers of patients. Future work will embrace more diverse populations and focus on integrating both common and rare variants (including single-nucleotide and copy-number changes) into a single genetic risk score at scale.

## Methods

### Cohort characteristics and genotyping

*University Hospital of Bordeaux albinism cohort.* Individuals with albinism were identified through the database of the University Hospital of Bordeaux Molecular Genetics Laboratory, France. This is a national reference laboratory that has been performing genetic testing for albinism since 2003 and has been receiving samples from individuals predominantly based in France (or French-administered overseas territories).

Information on the dermatological and ophthalmological phenotypes was available for all affected individuals and each of these cases had at least one of the key ocular features of albinism, i.e. nystagmus or absence of a foveal pit (prominent foveal hypoplasia). No pre-screening based on genotype was undertaken and only individuals who were not knowingly related were included.

Genetic testing, bioinformatic analyses, and clinical interpretation were performed as previously described[17,27]. Briefly, most participants had gene-panel testing of 19 genes associated with albinism (*TYR, OCA2, TYRP1, SLC45A2, SLC24A5, C10ORF11, GPR143, HPS 1 to 10, LYST, SLC38A8*) using IonTorrent platforms. High-resolution array-CGH (comparative genomic hybridisation) was also used to detect copy number variants in these 19 genes. All genetic changes of interest were confirmed with an alternative method (e.g. Sanger sequencing or quantitative PCR). Clinical interpretation of variants was performed using criteria consistent with the 2015 American College of Medical Genetics and Genomics best practice guidelines[44]. Generally, variants with MAF ≥ 1% in large publicly available datasets (e.g. gnomAD[28]) were considered unlikely to be disease-causing. We note that the genetic findings in a subset of this cohort (70%; 845/1208) have been partly reported in a previous publication by our group[17] (see Supplementary Data 1 for further information).

Due to the limited number of genes screened in this cohort, it was not possible to reliably assess genetic ancestry and to objectively assign individuals to ancestry groups. Attempting to mitigate this, we processed available data on self-identified ethnicity that were collected through questionnaires. Responses were inspected and stratification into five broad continental groups (European, African, Admixed American, East Asian, South Asian) was performed.

*Genomics England 100,000 Genomes Project cohort.* Clinical and genomic data from the Genomics England 100,000 Genomes Project were accessed through a secure Research Environment that is available to registered users. This dataset was collected as part of a national genome sequencing initiative[45]. Enrolment was coordinated by Genomics England Limited and participants were recruited mainly at National Health Service (NHS) Hospitals in the UK[7]. Clinical information was recorded in Human Phenotype Ontology (HPO)[46] terms and International Classification of Diseases (ICD) codes. Genome sequencing was performed in DNA samples from 78,195 individuals using Illumina HiSeq X systems (150 base-pair paired-end format). Reads were aligned using the iSAAC Aligner v03.16.02.19 and small variants were called using Starling v2.4.7[47]. Aggregation of single-sample gVCFs was performed using the Illumina software gVCF genotyper v2019.02.29; normalisation/decomposition was implemented by vt version 0.57721[48]. The multi-sample VCF was then split into 1371 roughly equal chunks to allow faster processing and the loci of interest were queried using bcftools v1.9[49] (see https://research-help.genomicsengland.co.uk/display/GERE/ for further information). Only variants that passed all provided site quality control criteria were processed. In addition, we filtered out genotypes with: genotype score <20; read depth <10; allele balance <0.2 and >0.8 for heterozygotes; allele balance >0.1 or <0.9 for homozygotes (reference and alternate, respectively). Genomic annotation was performed using Ensembl VEP[50]; one additional annotation was included—presence of a variant in HGMD v2021.2[40] with a 'disease-causing' (DM) label.

Ancestry inference was performed in this cohort using principal component analysis. Data from the 1000 Genomes Project (phase 3) dataset[45] were used and five broad super-populations were projected (European, African, Admixed American, East Asian, South Asian) (further information on this can be found online at https://research-help.genomicsengland.co.uk/display/GERE/Ancestry+inference).

We focused on a pre-determined subset of the Genomics England 100,000 Genomes Project dataset that includes only unrelated probands (*n* = 29,602). 105 of these individuals had a diagnosis of albinism, i.e. the ICD-10 term 'Albinism' [E70.3] and/or the HPO terms 'Albinism' [HP:0001022], 'Partial albinism' [HP:0007443] or 'Ocular albinism' [HP:0001107] were assigned. Together with the University Hospital of Bordeaux cases, these 105 probands formed the 'case' cohort (for the albinism risk analysis). The remaining 29,497 probands had no recorded diagnosis or phenotypic features of albinism and formed the 'control' cohort. We note that in-depth ophthalmic phenotyping was not routinely undertaken in Genomics England 100,000 Genomes Project participants. Thus, we cannot be

certain that a small number of individuals with mild/subclinical forms of albinism has not been included in the control cohort.

### Identifying functional regulatory and protein-coding variants

*Regulatory variants.* Focusing on *TYR*, we identified changes that are likely to have an impact on gene regulation by selecting variants that:

- are known *TYR* eQTLs.
- alter *TYR* cis-regulatory elements, including the promoter of the gene.

To identify eQTLs, we inspected the eQTL catalogue[51] and used data from the Genotype-Tissue Expression v8[52] and Eye Genotype Expression[53] projects. To identify regulatory elements, we used the ENCODE 3 (ENCyclopedia Of DNA Elements phase 3) dataset; the SCREEN (Search Candidate cis-Regulatory Elements by ENCODE) v10 interface was utilised to query this resource for regions flagged as candidate cis-regulatory elements (see https://screen.encodeproject.org/ for further information and definitions)[29]. Additional putative regulatory elements were identified by inspecting chromatin accessibility peaks in RPE samples in DESCARTES (the Developmental Single-Cell Atlas of Gene Regulation and Expression)[54] and through an extensive search of the biomedical literature (e.g.[30]). All these queries were conducted in January 2021.

*Common protein-coding variants.* Focusing on *TYR*, we identified common changes that are likely to have an impact on protein function by selecting variants that:

- have a CADD PHRED-scaled score ≥20. CADD is a widely-used integrative annotation tool built from more than 60 genomic features. A PHRED-scaled score ≥10 indicates a raw score in the top 10% of all possible single nucleotide variants, while a score ≥20 indicates a raw score in the top 1%[22]; it is noted that a cut-off of 20 has balanced sensitivity and specificity (90% and 69%, respectively)[55] in the context of this non-diagnostic setting.
- alter protein-coding sequences—including missense changes, nonsense variants and small insertions/deletions; variants with a potential role in splicing (e.g. synonymous changes and variants altering splice donor/acceptor sites) were not included.
- have 'total' MAF ≥ 1% in gnomAD v2.1.1[28].

*Rare protein-coding variants.* Focusing on 19 albinism-related genes (*TYR, OCA2, TYRP1, SLC45A2, SLC24A5, C10ORF11, GPR143, HPS 1 to 10, LYST, SLC38A8*), we identified rare changes that are likely to have an impact on protein function by selecting variants that:

- are labelled as disease-causing (DM) in HGMD v2021.2.
- are included in the following HGMD v2021.2 'mutation type' categories: missense/nonsense, splicing, small deletions, small insertions or small indels; gross deletions, gross insertions/duplications and complex rearrangements were not analysed.
- have 'total' MAF < 1% in gnomAD v2.1.1[28].

### Case-control analysis to estimate albinism risk

The effect of homozygosity for selected *TYR* haplotypes (formed by one common regulatory change, c.−301C>T, and two common protein-coding variants, c.575C>A and c.1205G>A) on albinism risk (i.e. the probability of receiving a diagnosis of albinism) was estimated using data from the University Hospital of Bordeaux albinism cohort and the Genomics England 100,000 Genomes Project dataset. A case–control analysis of a binary trait (presence/absence of albinism) was conducted assuming a recessive model. Logistic regression using the Firth bias reduction method[32,33] was utilised (as implemented in 'logistf' R package)[50]. The following covariates were included: gender, number of rare HGMD-listed variants and ancestry (Supplementary Table 2).

Although ancestry was included as a covariate in our logistic regression model, this analysis is imperfect as it was not possible to objectively determine genetic ancestry in the University Hospital of Bordeaux albinism cohort (as mentioned above, self-identified ethnicity was instead used as a surrogate). Confounding by ancestry (i.e. population stratification) is, therefore, a possibility. This can arise if there are systematic differences in ancestral background between cases and controls; these differences can result in significantly different allele and genotype frequencies between the compared groups which in turn may lead to spurious association signals[56]. We attempted to quantify the bias in our data by calculating the genomic inflation factor lambda (λGC). λGC is conceptually simple and involves using a set of random genetic markers to quantitatively estimate the structural differences between the case and control populations[42,43]. The selected λGC markers have to be un-linked and should not be expected to show an association with the trait under study (albinism or skin pigmentation in this case). We, therefore, selected 35 single-nucleotide variants that (i) had a CADD PHRED-scaled score <5 (i.e. are unlikely to be functionally relevant) and (ii) were genotyped both by the gene panels used in the University Hospital of Bordeaux albinism cohort and the genome sequencing assays used in Genomics England 100,000 Genomes Project (Supplementary Table 6). Subsequently, case–control comparisons were made

for each of these 35 λGC markers using Firth regression analysis. The resulting test statistics were then used to calculate the median value of λGC.

To further understand the impact of recent ancestry on our results, we analysed selected subsets of the case and control cohorts. First, we performed sub-analysis of the Genomics England 100,000 Genomes Project cases ($n = 105$) and controls ($n = 29,497$). Then we aimed to compare groups that were matched both in terms of genetic ancestry and geographical origin; thus we focused on the Genomics England 100,000 Genomes Project cases ($n = 76$) and controls ($n = 22,927$) that have European ancestries (as inferred by principal component analysis). Finally, we repeated our primary analysis using individuals from both the University Hospital of Bordeaux and the Genomics England 100,000 Genomes Project cohorts but this time focusing only on the cases ($n = 1107$) and controls ($n = 22,927$) that have European ancestries. The results are shown in Supplementary Fig. 3 and Supplementary Tables 7–9.

**Analysis of visual acuity and foveal thickness in UK Biobank participants**. The effect of homozygosity for selected *TYR* haplotypes was studied in UK Biobank participants. UK Biobank is a biomedical resource containing in-depth genetic and health information from >500,000 individuals from across the UK[6]. A subset of UK Biobank volunteers underwent enhanced phenotyping including visual acuity testing (131,985 individuals) and imaging of the central retina (84,748 individuals)[57]; the latter was obtained using optical coherence tomography (OCT), a non-invasive imaging test that rapidly generates cross-sectional retinal scans at micrometre-resolution[58]. All UK Biobank volunteers analysed as part of this study were imaged using the 3D OCT-1000 Mark II device (Topcon, Japan); the relevant methodology has been previously described[57]. Notably, only 24 UK Biobank participants are assigned a diagnosis of albinism (data field 41270; ICD-10 term 'Albinism' [E70.3]) of which only 7 had visual acuity measurements and none had OCT imaging; 19 additional individuals had a diagnosis of albinism in their primary care record data (resource 591). Given that reduced visual acuity and increased central retinal thickness (due to underdevelopment of the fovea) are two hallmark features of albinism, we decded to investigate the impact of *TYR* risk haplotypes on these quantitative endophenotypes.

First, genotyping array data were used to obtain genotypes for *TYR* c.575C>A [rs1042602] and *TYR* c.1205G>A [rs1126809] (data field 22418 including information from the Applied Biosystems UK Biobank Axiom Array containing 825,927 markers). In contrast to these two changes, the *TYR* c.−301C>T [rs4547091] variant was not directly captured by the array. However, high-quality (>99.9%) imputation data on this promoter change were available (data field 22828).

Subsequently, we calculated the mean of the right and left LogMAR visual acuity for each UK Biobank volunteer (data fields 5201 and 5208, 'instance 0' datasets). These visual acuity measurements were subsequently used to compare visual performance between groups of people with different homozygous haplotype combinations. As the obtained distributions deviated from normality (Fig. 2c), the Kruskal-Wallis test was used. Pair-wise comparisons were performed and the *p* values were adjusted using the Benjamini-Hochberg method (Supplementary Table 4).

To obtain central foveal thickness measurements from UK Biobank OCT images, we calculated the mean of the right and left retinal thickness (defined as the average distance between the hyperreflective bands corresponding to the RPE and the internal limiting membrane, across the central 1 mm diameter circle of the ETDRS grid) for each UK Biobank volunteer[51]. The obtained measurements were subsequently used to compare central macular thickness among groups of UK Biobank volunteers with different homozygous *TYR* haplotype combinations. As some of the obtained distributions deviated from normality (Supplementary Fig. 2), the Kruskal-Wallis test was used. Pair-wise comparisons were performed and the *p* values were adjusted using the Benjamini–Hochberg method (Supplementary Table 5).

**Ethics approval**. Informed consent was obtained from all participants or their parents in the case of minors. The study of individuals from the University Hospital of Bordeaux albinism cohort has been approved by the relevant local ethics committee (Comité de Protection des Personnes Sud-Ouest et Outre Mer III, Bordeaux, France). The informed consent process for the Genomics England 100,000 Genomes Project has been approved by the National Research Ethics Service Research Ethics Committee for East of England—Cambridge South Research Ethics Committee. The UK Biobank has received approval from the National Information Governance Board for Health and Social Care and the National Health Service North West Centre for Research Ethics Committee (Ref: 11/NW/0382). All investigations were conducted in accordance with the tenets of the Declaration of Helsinki.

**Reporting summary**. Further information on research design is available in the Nature Research Reporting Summary linked to this article.

## Data availability

Genomics England 100,000 Genomes Project data are available under restricted access through a procedure described at https://www.genomicsengland.co.uk/about-gecip/for-gecip-members/data-and-data-access. UK Biobank data are available under restricted access through a procedure described at http://www.ukbiobank.ac.uk/using-the-resource/. All other data supporting the findings of this study are available within the article (including its supplementary information files).

## Code availability

The scripts used to analyse the datasets included in this study are available at https://github.com/davidjohngreen/tyr.

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

## Acknowledgements

We acknowledge the following sources of funding: the Wellcome Trust (224643/Z/21/Z, Clinical Research Career Development Fellowship to P.I.S.; 200990/Z/16/Z, Transforming Genetic Medicine Initiative to G.C.B.); the UK National Institute for Health Research (NIHR) Clinical Lecturer Programme (CL-2017-06-001 to P.I.S.); Retina UK and Fight for Sight (GR586, RP Genome Project—UK Inherited Retinal Disease Consortium to G.C.B.); Christopher Green (D.J.G.); the French Albinism Association (Genespoir) and the French National Research Agency (Agence Nationale de la Recherche; ANR-21-CE17-0041-01 to B.A.). The UK Biobank Eye and Vision Consortium is supported by funding from the NIHR Biomedical Research Centre at Moorfields Eye Hospital and UCL Institute of Ophthalmology, the Alcon Foundation and the Desmond Foundation. The complete list of members of this Consortium can be found in the Supplementary Information. This research was made possible through access to the data and findings generated by the 100,000 Genomes Project. The 100,000 Genomes Project is managed by Genomics England Limited (a wholly-owned company of the Department of Health and Social Care). The 100,000 Genomes Project is funded by the NIHR and NHS England. The Wellcome Trust, Cancer Research UK and the Medical Research Council have also funded research infrastructure. The 100,000 Genomes Project uses data provided by patients and collected by the National Health Service (NHS) as part of their care and support. We acknowledge the contribution of the Genomics England Research Consortium to the 100,000 Genomes Project. The complete list of members of this Consortium can be found in the Supplementary Information. Lastly, we acknowledge the help of Cécile Courdier at the University Hospital of Bordeaux Molecular Genetics Laboratory, Jamie Ellingford at the University of Manchester, and Claire Hardcastle and Simon Ramsden at the North West of England Genomic Laboratory Hub.

## Author contributions

P.I.S. conceived and designed the experiments. The UK Biobank Eye and Vision Consortium, T.F., E.B., B.A., G.C.B. and P.I.S provided datasets and analytical tools. V.M., E.L., D.J.G., D.T.G., C.P., T.F., B.A. and P.I.S. analysed the data. P.I.S. wrote the manuscript with support from D.J.G. All authors critically revised and approved the manuscript. V.M., E.L. and D.J.G. contributed equally to this work as joint first authors; B.A. and G.C.B. contributed equally to this work as supporting principal investigators.

## Competing interests

E.B. is a paid consultant and equity holder of Oxford Nanopore, a paid consultant to Dovetail, and a non-executive director of Genomics England, a limited company wholly owned by the UK Department of Health and Social Care. All other authors declare no competing interests.

## Additional information

## UK Biobank Eye and Vision Consortium

Graeme C. Black [ID] [3,5 ✉] & Panagiotis I. Sergouniotis [ID] [3,4,5,6 ✉]

A full list of members and their affiliations appears in the Supplementary Information.

