## [Peer Review File · Nature Communications]

The contribution of common regulatory and protein-coding TYR variants in the genetic architecture of albinismREVIEWER COMMENTS

Reviewer #1 (Remarks to the Author):

REVIEW NCOMMS-21-41341-T

A Bordeaux-UK collaboration led to a very interesting study. The way the questioned their dataset is clever but some information is missing. In general it is unclear what was the population studied, clinical and molecular diagnosis. To be complete it would seem appropriate they bring an in vitro functional validation of their hypothesis. As for the extrapolation in the control population a bit more information was missing. How many had VA done and how many had OCT, those were all controls without albinism correct? I assume they excluded the cases identified as albinism? Did lesser vision correlate thicker retina. May show a graph of the distribution. More specific comments are below.

- 1) ABSTRACT: fourth line mention autosomal recessive from the start. Those were with a clinical diagnosis, were they all type 1 or all types of albinism. This should be clarified.
- 2) Were these cases with no molecular diagnosis and their hypothesis was that the regulation region was important. The why of their approach should be clarified or did they just look at anything they could find
- 3) Where are the cases recruited from, the 100000 genome project?
- 4) Is the homozygous haplotype acting like a hypomorphic variant.
- 5) Not sure what a risk predisposition for albinism is?

MAIN TEXT:

- 6) REMOVE 'for virtually all known rare disorders' that is an assumption
 - 7) SECOND PARAGRAPH: highlight at least type 1 and type 2 to give a context of importance
 - 8) The probands identified in the 100000 genomes project, were they clinically diagnosed with albinism or molecularly, what type. Again to give a context.
 - 9) The common variants, have they been previously associated with disease or presumed pathogenic?
 - 10) They are presumed functionally relevant.
 - 11) Line 92: suggest to remove; HOWEVER. Isn't that the definition of a hypomorph variant?
 - 12) LINE 97: to look into regulatory variants: they looked at eQTL and predicted enhancers? Right? Did they look at DNase hypersensitivity site and TFBS in the no-coding region or not? Just so it is clear what they look at.
 - 13) The haplotype bloc was arbitrarily defined To accept it they must make clearer that those three variants were the only one that met their hypothesized criteria and that they were all in albinism type 1 or not?
 - 14) Figure 2 b, did all cases have albinism? So what does the risk mean?
 - 15) Lines 131-137 on the risk of albinism. Where does the phenotype come from or is this a predictor. Was validated functionally. It would be nice to have info on the phenotype available and to summarize Suppl. table 1 as a pie chart or a graph. Though this is not the focus of their paper it is important information.
 - 16) For the association with VA and OCT that was in the control population, right? A difference of 0.1 logmar is within intertest variability. Were the OCTs done using the same type of instrument?
- Well done overall (as usual)

Reviewer #2 (Remarks to the Author):

The manuscript is a study of common and rare variation in the TYR gene and their association with albinism. The study looks at 1300 individuals with Albinism from an Albinism cohort and from the 100,000 genomes project. Other 100,000 genomes project individuals are used as controls. The effects of these TYR variants are also assessed in the UK Biobank against visual acuity. A haplotype of

common variants is found to associate with albinism and promoter variants are found to affect penetrance of rare pathogenic variants.

The paper is generally well written and the results are potentially interesting. I do have some major concerns though.

1. Population stratification seems like it could be a problem. It seems very important here for a trait like albinism where variants in the gene presumably have undergone selection. In fact the authors speculate that the promoter variant which varies in frequency from 80% to 40% in European vs. African ancestry may explain some differences in Albinism frequency. So clearly ancestry needs to be accounted for robustly.

But despite this the Bordeaux cohort which provides most of the cases (1200 out of 1300) could not infer genetic ancestry because of limited amounts of data - and self-reported ancestry was relied on. This is a major issue that seems hidden away in the methods section: "There was case-control imbalance and the two cohorts were imperfectly matched, especially in terms of genetic ancestry and genotyping approach used. These sources of bias should be taken into account when interpreting the results, especially findings with low-effect and/or low-confidence signal."

This might be less of a problem if the cases and controls were from the same study and had the same method of assigning ancestry, but using a case cohort from the South of France and using a control cohort from England without fully accounting for ethnicity differences seems very problematic. It's not just the ancestry differences, but as the authors seem to acknowledge, potential technical differences between cohorts. I would really like the authors to strongly justify why they think these major issues aren't affecting their conclusions. At the very least there should be a sub-analysis of just the 100,000 genomes cases and controls.

2. The authors say that a UK Biobank case/control analysis of Albinism wasn't possible because of a small number of ICD-10 cases. Have they looked to see how many cases are in the GP records - it may be the disease is better recorded there than in the in patient records? Using UK Biobank would get over the ancestry and technological difficulties.

3. Some of the criteria of why these variants were selected for analysis need expanding. For example, there's no justification of why only 1% with CADD>20 were analysed.

REVIEWER 1 COMMENTS

Comment #4: *A Bordeaux-UK collaboration led to a very interesting study. The way the questioned their dataset is clever but some information is missing. In general it is unclear what was the population studied, clinical and molecular diagnosis.*

Response #4: It is critical to provide clarity on the populations studied and we have now included further information on this. We have made changes to the main text and the methods, and have added the following to the legend of Figure 1 (see also responses #6, #8 and #10):

“A case-control analysis was performed to gain insights into the contribution of protein-coding and regulatory variation, at the common and rare ends of the allele-frequency spectrum, in albinism. The majority of participants in the ‘case’ cohort (1208/1313) were identified through the database of the University Hospital of Bordeaux Molecular Genetics Laboratory, France. All these probands had at least one key ocular feature of albinism, i.e. nystagmus or prominent foveal hypoplasia. The remaining 105/1313 cases were identified through the Genomics England 100,000 Genomes Project dataset and had a diagnosis of albinism or a phenotype deemed consistent with

partial/ocular albinism. The ‘control’ cohort included 29,497 unrelated individuals from the Genomics England 100,000 Genomes Project dataset, none of whom had a recorded diagnosis of albinism.”

Comment #5: *To be complete it would seem appropriate they bring an in vitro functional validation of their hypothesis.*

Response #5: As we discuss in P6 (L102-103) and P7 (L117-119), the three *TYR* variants that we studied (rs4547091, rs1042602 and rs1126809) have significant published functional data to support an impact on tyrosinase expression and/or activity (refs 29 and 30).

We agree that further *in vitro* experiments (e.g. involving patient-derived induced pluripotent stem cells) would be of interest and are likely to provide important insights into the link between RPE melanin synthesis and vision. To acknowledge this, we have now added the following (P11, L193-199): *“We expect that future studies analysing visual function and ocular structure in this group of homozygous individuals will provide key insights into the elusive link between RPE melanin synthesis and visual system organisation.³⁵ Furthermore, we anticipate that the study of cellular models specific to these homozygous cases (e.g. human induced pluripotent stem cell-derived RPE) will advance our understanding of the molecular pathology of albinism.”*

Nonetheless, we believe that our study provides a solid evidence basis and that our findings are robust, directly relevant to patients and, crucially, immediately clinically translatable.

Comment #6: *As for the extrapolation in the control population a bit more information was missing. How many had VA done and how many had OCT, those were all controls without albinism correct? I assume they excluded the cases identified as albinism?*

Response #6: We made every effort to remove albinism cases from the control population. The control cohort included 29,497 probands from the Genomics England 100,000 Genomes Project dataset (100K_GP). Individuals identified as having albinism (i.e 100K_GP participants that have been assigned the ICD-10 term “Albinism” and/or the HPO terms “Albinism”, “Partial albinism” or “Ocular albinism”) were excluded from this control group. This is discussed in the main text (P5, L89-92), the methods (P16, L319-323) and in Supplementary Table 2. As mentioned in response #4, we have now, for clarity, included further information on the case and control cohorts in the legend of Figure 1 (P5).

Regarding visual acuity and OCT findings: these are not available for 100K_GP participants and were therefore not available for individuals in our control cohort. Since these ophthalmic investigations can help to identify mild forms of albinism, we cannot be certain about the absence

of probands with subclinical albinism from the control cohort. We have now alluded to this in the methods (P16, L320-323):

“The remaining 29,497 probands had no recorded diagnosis or phenotypic features of albinism and formed the ‘control’ cohort. We note that in-depth ophthalmic phenotyping was not routinely undertaken in Genomics England 100,000 Genomes Project participants. Thus, we cannot be certain that a small number of individuals with mild/subclinical forms of albinism are not included in the control cohort.”

Comment #7: Did lesser vision correlate thicker retina. May show a graph of the distribution.

Response #7: There is no association between lesser vision and thicker central retina (see for example Poh et al. in Sci Rep 2020). Numerous confounders with various directions of effect are likely to be involved in this relationship. For example, people with macular oedema or foveal hypoplasia are likely to have reduced vision and thickened central retina while people with age-related macular degeneration (or other degenerative retinopathies) are likely to also have reduced vision but their central retina will be thinned. Adjusting for these confounding factors is challenging and although this analysis would be interesting, it is outside the scope of this study.

More specific comments are below.

Comment #8: ABSTRACT 1) fourth line mention autosomal recessive from the start. Those were with a clinical diagnosis, were they all type 1 or all types of albinism. This should be clarified.

Response #8: Individuals with all types of albinism (including autosomal recessive and X-linked recessive forms) were included in the case cohort. To minimise confusion, we have replaced “autosomal recessive” with “recessive” in the third sentence of the abstract (P3, L36-38) which now reads: *“Aiming to gain insights into the genetic architecture of rare recessive disorders, we studied a cohort of 1,313 individuals with albinism, an archetypal Mendelian condition.”*

For clarity, we added information on the characteristics of the case cohort in the legend of Figure 1 (P5): *“The majority of participants in the ‘case’ cohort (1208/1313) were identified through the database of the University Hospital of Bordeaux Molecular Genetics Laboratory, France. All these probands had at least one key ocular feature of albinism, i.e. nystagmus or prominent foveal hypoplasia. The remaining 105/1313 cases were identified through the Genomics England 100,000 Genomes Project dataset and had a diagnosis of albinism or a phenotype deemed consistent with partial/ocular albinism.”*

In addition to this, we expanded the relevant methods section and have included a new figure outlining the diagnoses in the University Hospital of Bordeaux cases (see responses #9 and #14 below).

Comment #9: ABSTRACT 2) *Were these cases with no molecular diagnosis and their hypothesis was that the regulation region was important. The why of their approach should be clarified or did they just look at anything they could find*

Response #9: Cases with or without a molecular diagnosis were included in the analysis. To clarify this, we have now added the following statement to the first paragraph of the methods section (P14, L256-257): *“No pre-screening based on genotype was undertaken”*

Regarding the ‘why’ of our approach: it is not possible to identify a molecular diagnosis in at least 1 in 4 probands with albinism (see P4, L76-78). We believe that rare protein-coding genetic variants tell only part of the story of genetic predisposition to albinism and we wanted to gain insights into the overall impact of genetic variation in this condition (including the into contribution of common regulatory and common protein-coding variants). We discuss this in the abstract (P3, L36-39) and in the first two paragraphs of the main text. We added another mention to this in the legend of Figure 1 (P5, *“A case-control analysis was performed to gain insights into the contribution of protein-coding and regulatory variation, at the common and rare ends of the allele-frequency spectrum, in albinism.”*)

Comment #10: ABSTRACT 3) *Where are the cases recruited from, the 100000 genome project?*

Response #10: Most cases (1208 out of 1313) were recruited through the University Hospital of Bordeaux Molecular Genetics Laboratory, France. The remaining cases (105 out of 1313) were recruited through the 100,000 genomes project. This is discussed in the third paragraph of the main text (P4-5, L83-90) and in the methods (P14-15). We attempted to clarify this further in the legend of Figure 1 (P5; see also responses #4 and #8).

Comment #11: ABSTRACT 4) *Is the homozygous haplotype acting like a hypomorphe variant.*

Response #11: We thank the reviewer for the opportunity to clarify the rationale behind the terminology that we are using/favouring. In rare/Mendelian disease literature, the term “hypomorphic” is used to describe variants that confer moderate risk of disease (when encountered in specific situations, e.g. in *cis* with a complete loss-of-function pathogenic change). In the common/complex disease literature, the term risk allele can be used to describe these changes (see Figure below). We believe that this distinction is somewhat artificial.

Genetic variant effects

In this study, we have shown that homozygosity for the *TYR* c.[-301C;575C>A;1205G>A] haplotype is a probabilistic marker for disease. One could describe this as a “hypomorphic genotype” or as a “risk genotype”. We have chosen the latter for two reasons: (i) the term hypomorphic is typically used to describe a variant, not a genotype, and we felt that, in this context, confusion might be caused; (ii) “hypomorphic” is usually a qualitative description; in contrast, “risk variants” are typically defined quantitatively and are generally characterised by their odds ratio. Notably, a key message from our study is that the quantitative, probabilistic approach underpinning common disease genetics can significantly enrich the study of rare/Mendelian disorders.

Comment #12: ABSTRACT 5) Not sure what a risk predisposition for albinism is?

Response #12: Risk/predisposition to disease is an intuitive concept in the context of acquired disorders. We agree however with the reviewer that this term can be confusing when one discusses congenital disease. This concept becomes clearer when risk to an embryo/foetus is considered. For clarity, we have now included definitions of the term ‘risk’: see P7, L136 in the main text, the legend of Figure 2 in P8, and P17, L373 in the Methods).

We have also modified the sentence in the abstract that might have caused confusion (P3, L42-46); this now reads: “*We also found that homozygosity for a haplotype formed by three common, functionally-relevant variants, TYR c.[-301C;575C>A;1205G>A], is associated with a high probability of receiving an albinism diagnosis (OR>82). This genotype is also associated with reduced visual acuity and increased central retinal thickness in UK Biobank participants.*”

Comment #13: MAIN TEXT 6) REMOVE ‘for virtually all known rare disorders’ that is an assumption

Response #13: As per reviewer comment, we have modified the sentence in P4, L64-66

from: “However, for virtually all known rare disorders, it is not possible to identify such pathogenic changes in every affected individual, leaving significant diagnostic and knowledge gaps.⁹⁻¹¹”
to “However, for most known rare disorders, it is not possible to identify such pathogenic changes in every affected individual, leaving significant diagnostic and knowledge gaps.⁹⁻¹¹”.

Comment #14: MAIN TEXT 7) SECOND PARAGRAPH: highlight at least type 1 and type 2 to give a context of importance

Response #14: We have highlighted the importance/prevalence of type 1 (TYR-related) and type 2 (OCA2-related) albinism in a new figure, Figure 3 (P10). This includes a break-down of the diagnoses that we obtained as part of our study and can be found below:

Figure 3. High level molecular diagnoses in 1208 probands from the University Hospital of Bordeaux albinism cohort.

(Figure legend):

It was not possible to detect a molecular diagnosis in 24% of cases (292/1208; “unknown” category). The following genes were implicated in the remaining probands: TYR (34%; 411/1208), OCA2 (19%; 229/1208), SLC45A2 (8%; 91/1208), GPR143 (5%; 65/1208), Hermansky-Pudlak syndrome (HPS) related genes (5%; 59/1208), SLC24A5 (2%; 28/1208), other albinism-related genes (3%; 33/1208). A significant subset of cases with TYR-related albinism were found/presumed to carry either the TYR c.[-301C;575A;1205A] or the TYR c.[-301C;575C;1205A] haplotype (15%, 187/1208, in heterozygous state [“TYR het haplotype” category]; 3%, 37/1208, in homozygous state [“TYR hom haplotype” category]). Further information including a list with all molecular diagnoses can be found in Supplementary Table 1.

We note that, as discussed in Supplementary Table 2, the importance/prevalence of specific albinism subtypes varies in different ancestry groups and a detailed discussion on this is beyond the scope of this manuscript.

Comment #15: MAIN TEXT 8) The probands identified in the 100000 genomes project, were they clinically diagnosed with albinism or molecularly, what type. Again to give a context.

Response #15: We have tried to address this in our response to comments #6, #8, #9 and #10. As mentioned above, the probands were clinically diagnosed and a broad definition of albinism (including ocular/partial forms) was used.

Comment #16: MAIN TEXT 9) The common variants, have they been previously associated with disease or presumed pathogenic?

Response #16: As of 14/02/2022, none of these *TYR* changes is labelled as pathogenic in ClinVar. More specifically:

- the c.-301C>T [rs4547091] promoter variant has not been reported in ClinVar.
- for c.575C>A (p.Ser192Tyr) [rs1042602], there are “conflicting interpretations of pathogenicity” (6 x Benign; 1 x Uncertain significance) [<https://www.ncbi.nlm.nih.gov/clinvar/variation/3778/>].
- for c.1205G>A (p.Arg402Gln) [rs1126809], there are also “conflicting interpretations of pathogenicity” (2 x Benign, 2 x Likely Benign, 3 x Uncertain significance, 1 x Likely pathogenic) [<https://www.ncbi.nlm.nih.gov/clinvar/variation/3779/>]. We attempted to summarise the most widely accepted view about the role of this variant in P6, L103-106: “*there is evidence suggesting that c.1205G>A is acting as a hypomorphic variant and is causing a mild form of albinism when in compound heterozygous state with a complete loss-of-function TYR mutation.*²⁶”.

Our study provides a much clearer picture of the role of these variants on albinism whilst highlighting the value of a haplotype-based approach.

Comment #17: MAIN TEXT 10) They are presumed functionally relevant.

Response #17: It is unclear to which section of the text this comment is referring to. The only mention to “functionally relevant” variants is in P6, L97-98 (“*are predicted by a computational algorithm to be functionally relevant (CADD score ≥ 20)*”). Adding “presumed” to this sentence would be a pleonasm.

Comment #18: MAIN TEXT 11) Line 92: suggest to remove; HOWEVER. Isn't that the definition of a hypomorph variant?

Response #18: We have now changed the relevant sentence (P6, L106-109)

from: “*However, 27% and multiple unaffected homozygous individuals have been reported (including >2,000 people in the control subset of the Genome Aggregation Database [gnomAD] v2.1.1)*²⁷.”

to: “*It is also noted that the MAF of this variant in European populations is around 27% and multiple unaffected homozygous individuals have been reported (including >2,000 people in the control subset of the Genome Aggregation Database [gnomAD] v2.1.1)*²⁷.”

Comment #19: MAIN TEXT 12) LINE 97: to look into regulatory variants: they looked at eQTL and predicted enhancers? Right? Did they look at DNase hypersensitivity site and TFBS in the non-coding region or not? Just so it is clear what they look at.

Response #19: As the reviewer states, we looked at eQTLs that alter predicted enhancers. Mentions to this are included in the main text (P6, L111-114) and the methods (P16, L327-330). We have expanded and clarified the methods section which now includes the following (P16, L332-341):

“To identify regulatory elements, we used the ENCODE 3 (ENCyclopedia Of DNA Elements phase 3) dataset; the SCREEN (Search Candidate cis-Regulatory Elements by ENCODE) v10 interface was utilised to query this resource for regions flagged as candidate cis-regulatory elements (see <https://screen.encodeproject.org/> for further information and definitions).²⁸ Additional putative regulatory elements were identified by inspecting chromatin accessibility peaks in RPE samples in DESCARTES (the Developmental Single Cell Atlas of Gene Regulation and Expression)⁴⁷ and through an extensive search of the biomedical literature (e.g.²⁹).”

Comment #20: MAIN TEXT 13) The haplotype bloc was arbitrarily defined To accept it they must make clearer that those three variants we the only one that met their hypothesized criteria and that they were all in albinism type 1 or not?

Response #20: Thank you for the opportunity to clarify any misperceptions on how the haplotype blocks were defined. Indeed the *TYR* gene (linked to albinism type 1) is the focus of our case-control analysis. First, aiming to identify variants of interest (i.e. functionally relevant changes), we set specific criteria for regulatory and common protein-coding variants. Three changes met these criteria and these form specific haplotype blocks. We are discussing this in the main text (e.g. P6, L95-98; P6, L111-114) and the in the methods. For clarity, we have now included another mention to this in the legend of Figure 2 (P5): *“Genotypes that include selected TYR haplotypes in homozygous state were studied. Haplotypes of interest were defined as those formed by combinations of TYR variants that are predicted to be functionally relevant; three variants met the pre-determined criteria set for regulatory (TYR c.-301C>T) or common protein-coding (TYR c.575C>A and c.1205G>A) variants. The associated haplotypic blocks were analysed further using logistic regression (Methods).”*

Comment #21: MAIN TEXT 14) Figure 2 b, did all cases have albinism? So what does the risk mean?

Response #21: As per previous responses, all cases had albinism and the overwhelming majority of the controls did not. We used a case-control analysis to estimate the risk in a hypothetical situation, e.g. of an embryo carrying the genotypes under study.

For clarity, we have now included the following in the figure legend: "*Risk of albinism (i.e. probability of receiving a diagnosis of albinism)*".

Comment #22: MAIN TEXT 15) Lines 131-137 on the risk of albinism. Where does the phenotype come from or is this a predictor. Was is validated functionally. It would be nice to have info on the phenotype available and to summarize Suppl. table 1 as a pie chart or a graph. Though this is not the focus of their paper it is important information.

Response #22: Indeed the risk of albinism refers to a prediction (see also responses #12 & #21). Functional validation is discussed in response #5. We have now summarised Supplementary Table 1 in a pie chart (Figure 3) (see response #14).

Comment #23: MAIN TEXT 16) For the association with VA and OCT that was in the control population, right? A difference of 0.1 logmar is within intertest variability. Were the OCTs done using the same type of instrument?

Response #23: We used the UK Biobank to study if the genotypes that are associated with an increased risk of albinism have an impact on key albinism 'endophenotypes' (VA and OCT-measured foveal thickness). We note that the UK Biobank is not considered a control population; although most participants do not have albinism, there are exceptions (P19, L427-430: "*only 24 UK Biobank participants are assigned a diagnosis of albinism of which only 7 had visual acuity measurements and none had OCT imaging*").

Regarding the type of instrument used: the same OCT device was used to image all UK Biobank participants analysed in this study. We have now included the following in the methods section P19, L426-427: "*All UK Biobank volunteers analysed as part of this study were imaged using the 3D OCT-1000 Mark II device (Topcon, Japan); the relevant methodology has been previously described.*⁵²".

REVIEWER 2 COMMENTS

Comment #24: 1. Population stratification seems like it could be a problem. This seems very important here for a trait like albinism where variants in the gene presumably have undergone selection. In fact the authors speculate that the promoter variant which varies in frequency from 80% to 40% in European vs. African ancestry may explain some differences in Albinism frequency. So clearly ancestry needs to be accounted for robustly.

But despite this the Bordeaux cohort which provides most of the cases (1200 out of 1300) could not infer genetic ancestry because of limited amounts of data - and self-reported ancestry was relied on. This is a major issue that seems hidden away in the methods section: "There was case-control imbalance and the two cohorts were imperfectly matched, especially in terms of genetic ancestry and genotyping approach used. These sources of bias should be taken into account when interpreting the results, especially findings with low-effect and/or low-confidence signal."

This might be less of a problem if the cases and controls were from the same study and had the same method of assigning ancestry, but using a case cohort from the South of France and using a control cohort from England without fully accounting for ethnicity differences seems very problematic. It's not just the ancestry differences, but as the authors seem to acknowledge, potential technical differences between cohorts. I would really like the authors to strongly justify why they think these major issues aren't affecting their conclusions. At the very least there should be a sub-analysis of just the 100,000 genomes cases and controls.

Response #24: We agree with the reviewer and acknowledge the major implications that ancestry effects can have (and how these can lead to confounding). The approach we used in the initial version, which involved a combination of data-assigned with self-reported ancestry, is imperfect and we have performed further analyses to address this issue. We used four different approaches and our new observations strengthen the results of the study.

1. First, following the reviewer's suggestion, we performed a sub-analysis of the Genomics England 100,000 Genomes Project (100K_GP) cases and controls.
2. We took this further and focused on 100K_GP cases and controls that have European ancestries (as inferred by principal component analysis).
3. We then repeated our primary analysis using individuals from both the Bordeaux and 100K_GP cohorts but this time focusing only on cases and controls that have European ancestries. Self-reported ancestry information was used for cases in the Bordeaux cohort. Principal component analysis was used to infer ancestry in 100K_GP cases and controls.

Notably, analyses 1, 2 and 3 revealed that the [C;A;A] and [C;C;A] haplotypes in homozygous state are associated with a significant increase in albinism risk. This is in keeping with the findings of our primary analysis. These observations are summarised in a new figure (Supplementary Figure 3):

Supplementary Figure 3. Secondary analyses in selected subsets of the case and control cohorts confirm that common TYR variants form haplotypes that affect albinism risk

(Figure legend): The x-axes are showing odds ratios (\log_{10}) and correspond to the probability of receiving a diagnosis of albinism in individuals who carry a specific haplotype in homozygous state.

Panel **a**, shows the results obtained following analysis of a mixed case cohort (including 1208 probands from the University Hospital of Bordeaux cohort and 105 cases from the 100K_GP dataset) and an inclusive control cohort (including 29,497 unrelated individuals from the 100K_GP dataset). This graph is identical to that shown in Figure 2b and is included here to facilitate comparison with the graphs in the other panels. Relevant numeric data can be found in Supplementary Table 3.

Panel **b**, shows the results obtained following a similar analysis to that in panel **a**, but this time only focusing on individuals with predominantly European ancestries. Further information and relevant numeric data can be found in Supplementary Table 10.

Panel **c** shows the results obtained following analysis of cases and controls from the 100K_GP dataset only. Further information and relevant numeric data can be found in Supplementary Table 8.

Panel **d** shows the results obtained following a similar analysis to that in panel **c**, but this time only focusing on individuals with predominantly European ancestries. Further information and relevant numeric data can be found in Supplementary Table 9.

Overall, the findings of the sub-analyses discussed in panels **b**, **c** and **d** are well aligned with those of the primary analysis shown in panel **a** (and Figure 2b)

[C;A;A] corresponds to homozygosity for TYR c.[-301C;575A;1205A];
 [C;A;G] corresponds to homozygosity for TYR c.[-301C;575A;1205G];

[T;C;A] corresponds to homozygosity for *TYR* c.[-301T;575C;1205A];
[C;C;A] corresponds to homozygosity for *TYR* c.[-301C;575C;1205A];
[T;C;G] corresponds to homozygosity for *TYR* c.[-301T;575C;1205G];
[C;C;G] corresponds to homozygosity for *TYR* c.[-301C;575C;1205G];
100K_GP corresponds to Genomics England 100,000 Genomes Project.

4. To understand the differences between the case and control cohorts in our primary analysis we calculated the genomic inflation factor lambda (λ GC). λ GC is an established method for assessing for lack of matching between cases and controls [Devlin and Roeder, *Biometrics* 1999]. Briefly, λ GC works by using a set of random, unlinked single nucleotide polymorphisms (SNPs) that have been genotyped to determine the inflation in the test statistic induced by population stratification. We selected a set of SNPs that were analysed both by the Bordeaux gene panels and the 100K_GP genome sequencing assays. Notably, the Bordeaux panels were designed to test pigmentation-associated loci and selecting truly neutral SNPs for λ GC analysis among the variants genotyped by the gene panel required additional consideration. We chose to focus on variants with a CADD score less than 5 and selected a set of 35 SNPs (Supplementary Table 7). We then used Firth regression analysis to estimate λ GC and found it to be 1.04; we note that the expected λ GC value is 1 so this observation increases confidence in the validity of the detected associations.

The relevant methodology for analyses 1-4 is discussed in a new methods section (P18, L388-417). The associated findings are discussed in Supplementary Figure 3 and Supplementary Tables 7-10.

Although the results of these additional analyses are encouraging and revealed no evidence for systematic bias, we have included the following in the manuscript's penultimate paragraph of the manuscript (P12, L219-233):

“One potential limitation of this study is our inability to stringently match the albinism cases with the unaffected controls, especially in terms of recent ancestry (which can be correlated with skin pigmentation). Although ancestry was included as a parameter in our regression model, this analysis was imperfect as it was not possible to reliably assign genetic ancestry to most albinism cases. It is known that inability to fully account for differences in ancestral background between cases and controls can lead to false-positive association signals.³⁷ We used a combination of orthogonal approaches to evaluate the robustness and generalisability of our findings. First, we used 35 presumed neutral single-nucleotide variants to calculate the genomic inflation factor lambda (λ GC)^{38,39}; λ_{median} was found to be 1.04, in keeping with limited confounding by ancestry (Supplementary Table 7). Subsequently, we performed targeted sub-analyses of the available cohorts; the results of three focused case-control studies supported our key findings and increased confidence in the validity of the detected associations (Supplementary Figure 3 and Supplementary Tables 8-10).”

Comment #25: 2. The authors say that a UK Biobank case/control analysis of Albinism wasn't possible because of a small number of ICD-10 cases. Have they looked to see how many cases are in the GP records - it may be the disease is better recorded there than in the in patient records? Using UK Biobank would get over the ancestry and technological difficulties.

Response #25: We did consider using the UK Biobank to obtain both cases and controls. However, even when including individuals with an albinism diagnosis provided by their primary care physician, the total number is only 43 (24 identified using the ICD codes provided in the "hesin_diag" tables and 19 identified using GP records); a mention to this has now been included in P19, L429-430. This observation suggests that the UK Biobank uses a mixture of diagnostic approaches/definitions and has an overall low number of cases. Since there are 105 albinism patients in the ~30,000 unrelated individuals from 100K_GP, we felt that this was a more appropriate dataset for our analysis. As mentioned in response #24, we have now re-analyzed the data using a subset of individuals from 100K_GP whose ancestry was calculated using principal component analysis. Despite an obvious reduction in statistical power, we observed similar signals, particularly for the high-risk haplotypes [C;A;A] and [C;C;A]. We feel that these findings, combined with the statistically significant reduction in visual acuity and retinal thickness observed in an entirely unrelated dataset (UK Biobank), provide additional support for our results.

Comment #26: 3. Some of the criteria of why these variants were selected for analysis need expanding. For example, there's no justification of why only 1% with CADD>20 were analysed.

Response #26: The contribution of rare genetic variants in the genetic architecture of albinism has been extensively studied in the past and here we wanted to gain insights into the role of common variants (see also response #9). Various minor allele frequency cut-offs have been previously used to distinguish common from rare variants with 1% being a commonly used threshold (see for example Li et al, Nature 2017 or Bloom et al. eLife 2019). Regarding CADD and the CADD score cut-off that we selected: CADD is a widely used variant pathogenicity prediction tool. It is regularly utilised by the clinical genetic laboratories linked to our groups and comparative studies have shown that it consistently outperforms many other *in silico* algorithms (see for example van der Velde et al, Genome Biol 2017, Li et al, Genome Med 2020 or Livesey et al, Mol Syst Biol. 2020). There is no single universal cut-off value for CADD scores. However, 20 is a commonly used threshold in the context of Mendelian disorders [Li et al, Genome Med 2020] with the median sensitivity and median specificity for this cut-off being 90% and 69% respectively (the values corresponding to a cut-off of 15 are 94% and 57%; those corresponding to a cut-off of 25 are 71% and 95%) [van der Velde et al, Genome Biol 2017]. Our intention was to identify likely deleterious variants and, for this purpose, we

felt that the balance between sensitivity and specificity offered by a cut-off of 20 was most appropriate. We have now included the following to the methods section (P17, L350-354):

“CADD is a widely-used integrative annotation tool built from more than 60 genomic features. A PHRED-scaled score ≥ 10 indicates a raw score in the top 10% of all possible single nucleotide variants, while a score ≥ 20 indicates a raw score in the top 1%²² it is noted that a cut-off of 20 has balanced sensitivity and specificity (90% and 69% respectively)²³ in the context of this non-diagnostic setting.”

We hope that these changes have addressed the reviewers' comments and that you will now be able to consider our manuscript as suitable for publication in *Nature Communications*.

Thank you for your help and time,

Yours faithfully,

Panos Sergouniotis, David Green, Vincent Michaud, Eulalie Lasseaux and co-authors

REVIEWERS' COMMENTS

Reviewer #2 (Remarks to the Author):

My comments have been adequately addressed.

Reviewer #3 (Remarks to the Author):

Note: This is a review of a revised manuscript. An original reviewer was not available for re-review, so this is a fresh reading for me. I have also reviewed the previous reviewer's comments.

In this clearly-written article, Michaud et al. explore the genetic architecture (protein coding and noncoding) of albinism, specifically oculocutaneous albinism, type 1. Using a case-control approach that leverages the molecular diagnostic lab in Bordeaux and the 100K Genomes Project in the UK, they note a high-frequency promoter variant modulates the penetrance of a prevalent, disease-associated missense variant. In addition, a haplotype formed by three common, functionally-relevant variants is associated with a high probability of receiving an albinism diagnosis. The bioinformatics pipeline seems standard. The response to the previous reviews seem reasonable and I think the expanded explanations they now give make the paper better. My comments are relatively minor and largely center on contextualization of their findings.

1. Figure 1. It would be good here to include the protein changes predicted to result from the two common coding variants. These are noted elsewhere in the text, but the figure is a nice summary of the pipeline.

2. To Comment #5, 11, 12, etc. of the previous review: It may be helpful to explicitly note two things. 1) Although patients with no tyrosinase activity (OCA1A) have a fairly consistent phenotype, other patients with albinism clearly fall along a phenotypic continuum with varying degrees of iris transillumination, fundus hypopigmentation, foveal hypoplasia, visual acuity and optic nerve decussation abnormalities. It is explicitly because of this continuum that this study is notable in my opinion. Although there are numerous references that could be cited, perhaps the van Genderen criteria paper (PMID: 30098354) is the most comprehensive. The fact that consensus criteria for phenotypic diagnosis is a question underlines this continuum. 2) R402Q has not only been studied at the genetic level but at the protein functional level (see PMID: 27775880). I think citing functional protein data strengthen the authors' arguments towards defining risk alleles/hypomorphic variants.

3. Gronskov et al (Scientific Reports, 2019) performed a haplotype analysis in their Scandinavian cohort and identified two extremely rare SNVs that include the p.S192Y, p.R402Q variants in the haplotype block. A sentence contextualizing the authors' findings in relation to this study would be helpful in the discussion. Similarly, a very recent study in an Amish cohort (Lin et al, npy Genomic Medicine, 2022) might also fit with this point of discussion.

REVIEWERS' COMMENTS/REQUESTS

Comment #5: Reviewer #2 (Remarks to the Author):

My comments have been adequately addressed.

Response #5: We would like to thank the reviewer for their help and for giving us the opportunity to comprehensively address the important issue of population stratification.

Comment #6: Reviewer 3 (Remarks to the Author):

Note: This is a review of a revised manuscript. An original reviewer was not available for re-review, so this is a fresh reading for me. I have also reviewed the previous reviewer's comments.

In this clearly-written article, Michaud et al. explore the genetic architecture (protein coding and noncoding) of albinism, specifically oculocutaneous albinism, type 1. Using a case-control approach that leverages the molecular diagnostic lab in Bordeaux and the 100K Genomes Project in the UK, they note a high-frequency promoter variant modulates the penetrance of a prevalent, disease-associated missense variant. In addition, a haplotype formed by three common, functionally-relevant variants is associated with a high probability of receiving an albinism diagnosis. The bioinformatics pipeline seems standard. The response to the previous reviews seem reasonable and I think the expanded explanations they now give make the paper better. My comments are relatively minor and largely center on contextualization of their findings.

1. Figure 1. It would be good here to include the protein changes predicted to result from the two common coding variants. These are noted elsewhere in the text, but the figure is a nice summary of the pipeline

Response #6: We have now included the protein changes in Figure 1.

A mention to the protein-level alteration has also been included in the abstract.

Comment #7: 2. To Comment #5, 11, 12, etc. of the previous review: It may be helpful to explicitly note two things. 1) Although patients with no tyrosinase activity (OCA1A) have a fairly consistent phenotype, other patients with albinism clearly fall along a phenotypic continuum with varying degrees of iris transillumination, fundus hypopigmentation, foveal hypoplasia, visual acuity and optic nerve decussation abnormalities. It is explicitly because of this continuum that this study is notable in my opinion. Although there are numerous references that could be cited, perhaps the van Genderen criteria paper (PMID: 30098354) is the most comprehensive. The fact that consensus criteria for phenotypic diagnosis is a question underlines this continuum.

Response #7: Further to this helpful comment, we have now expanded the second paragraph of the Introduction which now includes the following: “*Although individuals with no residual tyrosinase activity have a consistent phenotype with visual impairment and near-total loss of melanin pigment in their eyes, skin and hair, most individuals with albinism fall along a phenotypic continuum with varying degrees of ocular and, often, cutaneous hypopigmentation.*”^[PMID: 30098354] *Building on recent work*^{17,21}, we sought to increase our understanding of the genetic complexity and clinical heterogeneity of this archetypal disorder.”

Comment #8: 2) R402Q has not only been studied at the genetic level but at the protein functional level (see PMID: 27775880). I think citing functional protein data strengthen the authors’ arguments towards defining risk alleles/hypomorphic variants.

Response #8: We have now included a reference to this key paper in the second paragraph of the Results and Discussion section.

Comment #9: 3. Gronskov et al (Scientific Reports, 2019) performed a haplotype analysis in their Scandinavian cohort and identified two extremely rare SNVs that include the p.S192Y, p.R402Q variants in the haplotype block. A sentence contextualizing the authors’ findings in relation to this study would be helpful in the discussion. Similarly, a very recent study in an Amish cohort (Lin et al, npy Genomic Medicine, 2022) might also fit with this point of discussion.

Response #9: We have now added a sentence that includes a reference to these studies (page 8): “*This conclusion is supported by the findings of three smaller-scale studies that used family-based methods and investigated haplotypes containing the TYR c.575C>A and c.1205G>A variants.*”^[Gronskov 2019; Campbell 2019; Lin 2022]

We note that our paper puts a spotlight on the *TYR* c.-301C>T variant which has been previously overlooked but appears to have a significant impact *in vivo*.